# MULTI-VIEW INDEPENDENT COMPONENT ANALYSIS WITH SHARED AND INDIVIDUAL SOURCES

## ABSTRACT

Independent component analysis (ICA) is a blind source separation method for linear disentanglement of independent latent sources from observed data. We investigate the special setting of noisy linear ICA where the observations are split among different views, each receiving a mixture of shared and individual sources. We prove that the corresponding linear structure is identifiable, and the shared sources can be recovered, provided that sufficiently many diverse views and data points are available. To computationally estimate the sources, we optimize a constrained form of the joint log-likelihood of the observed data among all views. We show empirically that our objective recovers the sources in high dimensional settings, also in the case when the measurements are corrupted by noise. Finally, we apply the proposed model in a challenging real-life application, where the estimated shared sources from two large transcriptome datasets (observed data) provided by two different labs (two different views) lead to a more plausible representation of the underlying graph structure than existing baselines.

## 1 INTRODUCTION

We consider a linear multi-view blind source separation (BSS) problem in the context of independent component analysis (ICA) where the different views share latent sources but also have view-specific ones. The modeling strategy presented in this work is inspired by applications from the biomedical domain where linear BSS problems have often been encountered due to the nature of the data (Vigário et al., 1997; McKeown & Sejnowski, 1998; Sompairac et al., 2019).

Linear multi-view BSS solutions, such as Group ICA (Calhoun et al., 2001), independent vector analysis (IVA) and its corresponding variations (Lee et al., 2008; Anderson et al., 2011; 2014; Engberg et al., 2016; Vía et al., 2011), have been widely used for analyzing fMRI and EEG data. Typical applications include multi-subject studies for unraveling group-level brain activity patterns in the data (Salman et al., 2019; Huster et al., 2015; Congedo et al., 2010; Durieux & Wilderjans, 2019; Congedo et al., 2010). The main assumption in all those models is that each view (e.g., subject data) is a linear mixture of brain activity patterns shared across views (e.g., the group-level brain activity pattern). However, there is a growing tendency in neuroscience to shift the focus from group-level inference to extracting individual-specific signals (Dubois & Adolphs, 2016). For instance, one can be interested in investigating the individual (individual-specific brain functions) and shared (behavioral phenotypes) patterns in individuals' brain activity in a natural stimuli experiment (Seghier & Price, 2018; Bartolomeo et al., 2017; Dubois & Adolphs, 2016). Unfortunately, the aforementioned multi-view methods cannot be directly applied in this case, unless they are part of a two step procedure (e.g. (Long et al., 2020)): applying ICA/IVA on the different views followed by statistical analysis to separate the individual from the shared sources.

Moreover, this particular multi-view task is not only relevant for neuroscience but also for computational biology, where ICA has been a standard tool for analyzing omics data (Sompairac et al., 2019; Avila Cobos et al., 2018; Fraunhoffer et al., 2022). For example, in cancer research, one assumes that the observed measurements of tissue/tumor samples are a linear mixture of cell-type specific expressions (latent sources) (Avila Cobos et al., 2018). Now, consider the task where we want to aggregate heterogeneous experimental datasets (e.g., from different labs or modalities) to improve cancer prediction. By utilizing the prior knowledge that the datasets have shared and experiment-

specific information, we can transform this data integration task into a linear multi-view BSS as the one discussed before.

**Model Summary.** To address this and similar applications, we formalize the stated BSS model as a linear noisy generative model for a multi-view data regime, assuming that the mixing matrix and number of individual sources are view-specific. By requiring that the sources are non-Gaussian and mutually independent, and the linear mixing matrices are with full column rank we provide identifiability guarantees for the mixing matrices and latent sources. We adopt a maximum likelihood approach for the joint log-likelihood of the observed views, which we use to estimate the mixing matrices. Furthermore, we suggest a novel strategy for data integration of transcriptome datasets. We show empirically that our method works well compared to the baseline methods when the estimated components are used for a graph inference task.

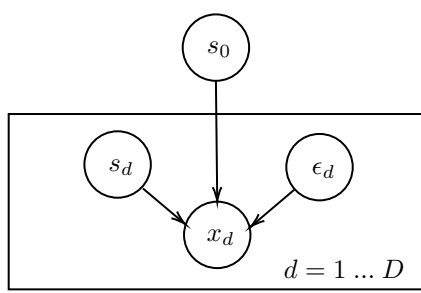

Figure 1: A graphical representation of 1

**Contributions.** Our contribution can be summarized as follows:

1. We provide theoretical guarantees for the identifiability of the recovered linear structure and the sources and noise distributions.

2. We propose an optimization procedure based on MLE for estimating the mixing matrices.

## 2 RELATED WORK

The existing body of work on linear multi-view BSS, inspired by the ICA literature, considers mostly shared response model applications (i.e., no individual sources), some of them adopting a maximum likelihood approach (Guo & Pagnoni, 2008; Richard et al., 2020; 2021) to model the noisy views of the proposed models. Many of these approaches, such as Group ICA (Calhoun et al., 2001), shared response ICA (SR-ICA) (Zhang et al., 2016), and MultiViewICA, incorporate a dimensionality reduction step for every view (CCA (Varoquaux et al., 2009; Richard et al., 2021) or PCA) to extract the mutual signal between the multiple objects before applying an ICA procedure on the reduced data. However, there are no guarantees that the pre-processing procedure will entirely remove the influence of the object-specific sources on the transformed data. Moreover, independent vector analysis (IVA) relaxes the assumption about the shared sources across views by allowing them to be dependent (Lee et al., 2008; Anderson et al., 2011; 2014; Engberg et al., 2016; Vía et al., 2011). However, as the previously discussed models, they fix the number of latent sources to be equal across views. To the best of our knowledge, two existing ICA-based methods exist for extracting shared and individual sources from data. Maneshi et al. (2016) proposes a heuristic way of using FastICA for the given task without discussing the identifiability of the results. The other exploits temporal correlations (Lukic et al., 2002) rather than the non-Gaussianity of the sources and thus is not applicable in the context we are considering.

A common tool for analyzing multi-view data is canonical correlation analysis (CCA), initially proposed by Hotelling (1936). It finds two datasets' projections that maximize the correlation between the projected variables. Gaussian-CCA (Bach & Jordan, 2005), its kernelized version (Bach & Jordan, 2002) and deep learning (Andrew et al., 2013) formulations of the classical CCA problem aim to recover shared latent sources of variations from the multiple views. There are extensions of CCA that model the observed variables as a linear combination of group-specific and dataset-specific latent variables. (Klami et al., 2013) estimated with Bayesian inference methods or exponential families with MCMC inference (Virtanen, 2010). However, most of them assume that the latent sources are Gaussian or non-linearly related to the observed data (Wang et al., 2016) and thus lack identifiability results.

Existing non-linear multiview versions such as (Tian et al., 2020; Federici et al., 2020) cannot recover both shared and individual signals across multiple measurements, and do not assure the identifiability of the proposed generative models. There are identifiable deep non-linear versions of

ICA which can be employed for this task. However, they make stronger assumptions to achieve identifiability which are hard to be satisfied in real-life applications (e.g. (Hyvärinen et al., 2019)).

## 3 PROBLEM FORMALIZATION

Consider the following $D$-view multivariate linear BSS model

$$x_d = A_d(\tilde{s}_d + \epsilon_d) = A_{d0}s_0 + A_{d1}s_d + A_d\epsilon_d, \qquad d \in \{1, \ldots, D\}, \tag{1}$$

where we assume that for $d = 1, \ldots, D$

- $x_d \in \mathbb{R}^{k_d}$ is a random vector with $\mathbb{E}[x_d] = 0$,
- $\tilde{s}_d = (s_0^\top, s_d^\top)^\top$ are latent random sources with and $s_0 \in \mathbb{R}^c$ and $s_d \in \mathbb{R}^{k_d - c}$ being the shared and individual sources and $\mathbb{E}[\tilde{s}_d] = 0$ and $\mathrm{Var}[\tilde{s}_d] = \mathbb{I}_{k_d}$,
- $A_d \in \mathbb{R}^{k_d \times k_d}$ is a mixing matrix with full column rank, $A_{d0}$ and $A_{d1}$ are the columns corresponding to the shared and individual sources,
- $\epsilon_d \sim \mathcal{N}(0, \sigma^2 \mathbb{I}_{k_d})$ is Gaussian noise, or measurement error, on the sources (similar to (Richard et al., 2020; 2021)).

Additionally, we assume that the variables $s_{01}, \ldots, s_{0c}, s_{11}, \ldots, s_{1(k_1-c)}, \ldots, s_{D1}, \ldots, s_{D(k_D-c)}$, $\epsilon_{11}, \ldots, \epsilon_{1k_1}, \ldots, \epsilon_{D1}, \ldots, \epsilon_{Dk_D}$ are mutually independent. Thus, we require that the noise variables do not influence the latent signal and vice versa. In this work, we aim to estimate the mixing matrices $A_d$ from given observations of $x_d, d = 1, \ldots, D$. Note that for $D = 1$ the model becomes a standard linear BSS model with non-Gaussian latent sources $z := \tilde{s}_1 + \epsilon_1$ (see (Comon, 1994; Hyvärinen & Oja, 2000; Bell & Sejnowski, 1995)).

## 4 IDENTIFIABILITY RESULTS

In unsupervised machine learning methods, the reliability of the algorithm cannot be directly verified outside of simulations due to the non-existence of labels. For this reason, theoretical guarantees are necessary to trust that the algorithm estimates the quantities of interest. For a BSS problem solution, such as ICA, we want the sources and mixing matrices to be (up to certain equivalence relations) unambiguously determined (or *identifiable*) by the data, at least in the large sample limit.

Identifiability results for noiseless single-view ICA are proved by (Comon, 1994). It turns out that if at most one of the latent sources is normal and the mixing matrix is invertible, then both the mixing matrix and sources can be recovered almost surely up to permutation, sign and scaling. However, this result does not hold in the general additive noise setting. Davies (2004) shows that if the mixing matrix has a full column rank, then the structure is identifiable, but not the latent sources. In the following we extend Comon (1994); Davies (2004); Kagan et al. (1973) results for the noisy setting inspired from our model (see 1). Compared to previous work, we provide identifiability guarantees for the sources and noise distributions.

**Theorem 4.1.** *Let $x_1, \ldots, x_D$ for $D \geq 2$ be random vectors with the following two representations:*

$$A_d^{(1)}\left(\begin{bmatrix} s_0^{(1)} \\ s_d^{(1)} \end{bmatrix} + \epsilon_d^{(1)}\right) = x_d = A_d^{(2)}\left(\begin{bmatrix} s_0^{(2)} \\ s_d^{(2)} \end{bmatrix} + \epsilon_d^{(2)}\right), \qquad d \in \{1, \ldots, D\},$$

*with the following properties for $i = 1, 2$*

1. *$A_d^{(i)} \in \mathbb{R}^{p_d \times k_d^{(i)}}$ is a (non-random) matrix with full column rank, i.e. $\mathrm{rank}(A_d^{(i)}) = k_d^{(i)}$,*

2. *$\epsilon_d^{(i)} \in \mathbb{R}^{k_d^{(i)}}$ and $\epsilon_d^{(i)} \sim \mathcal{N}(0, \sigma_d^{(i)2}\mathbb{I}_{k_d^{(i)}})$ is a $k_d^{(i)}$-variate normal random variable,*

3. *$\tilde{s}_d^{(i)} = (s_0^{(i)\top}, s_d^{(i)\top})^\top$ with $s_0^{(i)} \in \mathbb{R}^{c^{(i)}}$ and $s_d^{(i)} \in \mathbb{R}^{k_d^{(i)} - c^{(i)}}$ is a random vector such that:*

(a) the components of $\tilde{s}_d^{(i)}$ are mutually independent and each of them is a.s. a non-constant random variable,

(b) $\tilde{s}_d^{(i)}$ is non-normal with $0$ mean and unit variance.

4. $\epsilon_d^{(i)}$ is independent from $s_0^{(i)}$ and $s_d^{(i)}$: $\epsilon_d^{(i)} \perp\!\!\!\perp s_0^{(i)}$ and $\epsilon_d^{(i)} \perp\!\!\!\perp s_d^{(i)}$.

Then, $c^{(1)} = c^{(2)} =: c$ and for all $d = 1, \ldots, D$ we get that $k_d^{(1)} = k_d^{(2)} =: k_d$, and there exist a sign matrix $\Gamma_d$ and a permutation matrix $P_d \in \mathbb{R}^{k_d \times k_d}$ such that:

$$A_d^{(2)} = A_d^{(1)} P_d \Gamma_d,$$

and furthermore the sources and noise distributions are identifiable, i.e.

$$\begin{bmatrix} s_0^{(2)} \\ s_d^{(2)} \end{bmatrix} \sim \Gamma_d^{-1} P_d^{-1} \begin{bmatrix} s_0^{(1)} \\ s_d^{(1)} \end{bmatrix}, \qquad\qquad \sigma_d^{(2)} = \sigma_d^{(1)}.$$

Theorem 4.1, proved in Appendix A, assures the identifiability of the mixing matrices and *sources and noise distributions* up to sign and permutation for a multi-view ($D \geq 2$) noisy ICA model. This is a more general case than 1 since here the noise distribution can be view-specific and the mixing matrices can be non-square. We also provide identifiability results for the a more general single-view setting (see Theorem A.2). In the single-view case compared to the multiview one, we get similar but weaker results, i.e. the source and noise distributions are identifiable only if the latent sources do not have normal components. This means that for any of the latent sources variables $j$ in both equivalent representations in Theorem A.2, if we have $\tilde{s}_j^{(i)} \sim v + w$ with $v \perp\!\!\!\perp w$, then $v$ and $w$ are non-normal. Recent works by Richard et al. (2020; 2021); Anderson et al. (2014) provide identifiability results for a shared response modeling by imposing different assumptions to the one in Theorem 4.1. For example, under additional assumptions about the noise covariance matrices, the requirement about the non-Gaussianity of the sources in the ShICA model can be relaxed (Richard et al., 2021). Similar results about the shared sources are provided in Theorem A.4 by imposing additional requirements about the sources variance that are not covered by our model assumptions.

## 5 OPTIMIZATION

Here, we derive the joint log-likelihood of the observed views which we use for estimating the mixing matrices. Let $z_d := W_d x_d = \tilde{s}_d + \epsilon_d$, and $z_d^{(1)} := s_0 + \epsilon_{d0} \in \mathbb{R}^c$ and $z_d^{(2)} := s_d + \epsilon_{d1} \in \mathbb{R}^{k_d - c}$, i.e. $z_d = (z_d^{(1)\top}, z_d^{(2)\top})^\top$. Furthermore, let $p_{Z_d^{(2)}}$ be the probability distribution of $z_d^{(2)}$ and $|W_d| = |\det W_d|$. Then the data log-likelihood of 1 for $N$ observed samples per view is given by

$$\mathcal{L}(W_1, \ldots, W_D) = \sum_{i=1}^{N} \log f(\bar{s}_0^i) + \sum_{i=1}^{N} \sum_{d=1}^{D} \log p_{Z_d^{(2)}}(z_d^{(2)i}) + N \sum_{d=1}^{D} \log |W_d| \qquad (2)$$

$$- \frac{1}{2\sigma^2} \Big( \sum_{d=1}^{D} \text{trace}(Z_d^{(1)} Z_d^{(1)\top}) - \frac{1}{D} \sum_{d=1}^{D} \sum_{l=1}^{D} \text{trace}(Z_d^{(1)} Z_l^{(1)\top}) \Big) + C$$

where $Z_d^{(1)} \in \mathbb{R}^{c \times N}$ for $d = 1, \ldots, D$ is the data matrix that stores $N$ observations of $z_d^{(1)}$ and $\bar{s}_0^i = \sum_{d=1}^{D} z_d^{(1)i}/D$ and $f(\bar{s}_0) = \int \exp\Big( -\frac{D\|s_0 - \bar{s}_0\|^2}{2\sigma^2} \Big) p_{S_0}(s_0) ds_0$.

We further simplify the loss function by assuming that the data matrices $X_1 \in \mathbb{R}^{k_1 \times N}, \ldots, X_D \in \mathbb{R}^{k_D \times N}$ are whitened. That consists of centering and linearly transforming the random variables' realizations $x_d$ such that the resulting variable $\tilde{x}_d = K_d x_d$ has unit variance, $\mathbb{E}[\tilde{x}_d \tilde{x}_d^\top] = \mathbb{I}_{k_d}$, where $K_d$ is the whitening matrix. Thus, from the last equation we get that $\mathbb{I}_{k_d} = \mathbb{E}[\tilde{x}_d \tilde{x}_d^\top] = (1 + \sigma^2) K_d A_d A_d^\top K_d^\top$. It follows that the matrix $(1 + \sigma^2)^{\frac{1}{2}} K_d A_d$ is orthogonal, which we estimate by the matrix $W_d$. After training we set $\hat{A}_d = K_d^{-1} W_d$ which differs from the true one by $(1 + \sigma^2)^{\frac{1}{2}}$.

Due to the orthogonal constraints the objective function becomes

$$\mathcal{L}(W_1, \ldots, W_D) \propto \sum_{i=1}^{N} \log f_\sigma(\bar{s}_0^i) + \sum_{i=1}^{N} \sum_{d=1}^{D} \log p_{Z_d^{(2)}}(z_d^{(2)i}) + \frac{1+\sigma^2}{2D\sigma^2} \sum_{d=1}^{D} \sum_{l=1}^{D} \text{trace}(Z_d^{(1)} Z_l^{(1)\top})$$

(3)

where here $f_\sigma(\bar{s}_0) = \int \exp\left(-\frac{D\|s_0 - (1+\sigma^2)^{\frac{1}{2}}\bar{s}_0\|^2}{2\sigma^2}\right) p_{S_0}(s_0) ds_0$. This results from the fact that after whitening we have $\text{trace}(Z_d^{(1)} Z_d^{(1)\top}) = c$ and $|W_d| = 1$. All proofs can be found in Appendix B; the proof of equation 2 extends the one in (Richard et al., 2020). Note that in our optimization procedure, both $f_\sigma(\bar{s}_0)$ and $p_{Z_d^{(2)}}$, we approximate by the negative of a nonlinear function $g(s)$, e.g. $g(s) = \log\cosh(s)$ for super-Gaussian or $g(s) = -e^{-s^2/2}$ for sub-Gaussian sources. Moreover, since we do not estimate $\sigma^2$ we treat it as a Lagrange multiplier via the relation $\lambda = \frac{1+\sigma^2}{\sigma^2}$.

For the parameter estimation of the orthogonal unmixing matrices, we use the transformation framework proposed in (Lezcano-Casado, 2019; Lezcano-Casado & Martınez-Rubio, 2019)). The framework developed in (Lezcano-Casado, 2019) allows us to transform manifold optimization problems to unconstrained (Euclidean) optimization problems. To accomplish the transformation, the scheme uses trivializations, $\phi : \mathcal{V} \to \mathcal{M}$, which are smooth, surjective mappings between Euclidean spaces (e.g. $\mathcal{V}$) to the manifold (denoted by $\mathcal{M}$) (Lezcano-Casado, 2019). Thus, the optimization problem defined in 3 $\max_{W_1, \ldots, W_d \in \mathcal{M}} \mathcal{L}(W_1, \ldots, W_D)$ becomes equivalent to $\max_{y_1, \ldots, y_D \in \mathcal{V}} \mathcal{L}(\phi(y_1), \ldots, \phi(y_D))$ for $\mathcal{M}$ being the Stiefel manifold and $\phi$ a trivialization as the one described above. Subsequently, we can apply L-BFGS or (stochastic) gradient descent to compute approximate minimizers to our parameter estimation problem. Note that an alternative optimization approach is Riemannian optimization (Sato, 2021; Boumal et al., 2014). Moreover, there are other L-BFGS based methods, such as Ablin et al. (2018) that develop optimization solver for ICA under orthogonal constraints which relies on the Lie exponential map parametrization.

## 6 EXPERIMENTS

**Preprocessing.** Before running any of the ICA-based methods (our or the baselines) we whiten each single view by performing PCA to speed up computation.

**Model Implementation.** We used the python library `pytorch` (Paszke et al., 2017) to implement our method. We model each view with a separate unmixing matrix. To impose orthogonality constraints on the unmixing matrices, we made use of the `geotorch` library, which is an the extension of `pytorch` (Lezcano-Casado, 2019). The stochastic gradient based method applied for training is L-BFGS.

**Initialization.** We estimate the mixing matrix up to scale (due to the whitening) and permutation (see Sections 4 and 5). To force the algorithm to output the shared sources in the same order across all views we initialize the unmixing matrices by means of CCA. This follows from the fact that the CCA weights are orthogonal matrices due to whitening, and the transformed views' components are paired and ordered across views such that the first pair correspond to the largest singular value from the singular value decomposition of the cross correlation matrix.

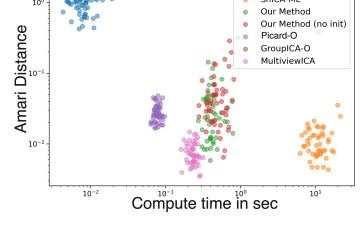

Figure 2: Comparison of the computing time (x-axis) and Amari distance (y-axis). The data is generated according to a shared response model with 5 views, 5 sources and 1000 samples and noise with the variance $\sigma = 1$. The addition "O" to the model's name refers to learning with orthogonal constraints.

**Training.** For all conducted experiments we fixed the parameter $\lambda$ from equation 3 to 1. In the simulated data experiments we conducted each experiment 50 times and based on that we provided error bars in all figures where applicable. For the computational specifics of the real-life data experiment please refer to Section 6.2.

**Baselines Implementation.** We compare our method to the standard single-view ICA method Infomax (`picard` library (Ablin et al., 2018)). To adopt it to the multi-view setting, we run Infomax

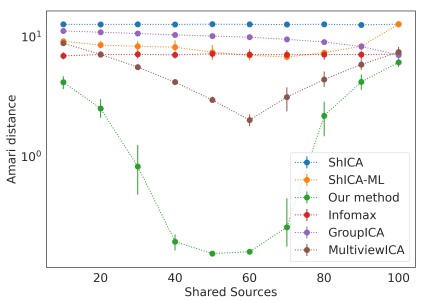
(a) Noiseless views according to 1

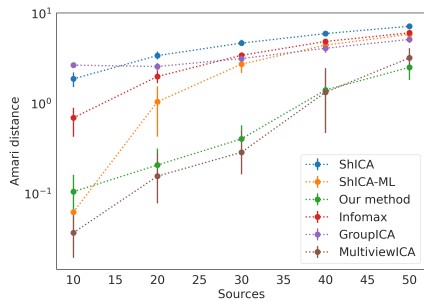
(b) Noisy views in shared response model

Figure 3: We compare the performance of our method, ShICA, Infomax, GroupICA, MultiViewICA and ShICA-ML Amari distance (the lower the better) for different number of shared sources (x-axis). The error bars correspond to 95% confidence intervals based on 50 independent runs of the experiment. Figure 3a: The datasets come from two different views with total number of sources 100 and sample size 1000. We vary the number of shared sources from 10 to 100. Figure 3b: The data is generated according to a model, where no individual sources are present and the noise per view is uniformly sampled from the interval $[1, 2]$. The number of views is set to 10 and the sample size is 1000. We vary the number of sources from 10 to 50 (y-axis).

on each view separately and then we apply the Hungarian algorithm (Kuhn & Yaw, 1955) to match components from different views based on their cross-correlation.

For the shared response model settings, we compare it to related methods such as MultiViewICA Richard et al. (2020), ShiCA, ShICA-ML Richard et al. (2021), and GroupICA as proposed by Richard et al. (2020). The latter involves a two-step pre-processing procedure, first whitening the data in the single views and then dimensionality reduction on the joint views. The code for the latter models is based on `https://github.com/hugorichard/multiviewica` (Richard et al., 2020) and for ShICA-ML `https://github.com/hugorichard/ShICA` (Richard et al., 2021).

For the data integration experiment we use a method based on partial least squares estimation, closely related to CCA, that extracts between-views correlated components and view-specific ones. This method is provided by the `OmicsPLS` R package Bouhaddani et al. (2018) and is especially developed for data integration task for omics data. We refer to this method as PLS. We also make use of IVA-L-SOS (Bhinge et al., 2019) as a representative of the independent vector analysis methods which aligns well with this task. This method assumes a linear noiseless model $x_d = A_d s_d, d \in \{1, \ldots, D\}$ with Laplacian independent sources $s_d \in \mathbb{R}^k$ per view where the following between view dependence holds: $(s_{1i}, \ldots, s_{Di}) \sim \text{Laplace}(0, \Sigma)$ with $s_{di}$ being the $i-$th component of $s_d$. The implementation of this method is provided by the `independent-vector-analysis` package provided by (Lehmann et al., 2022).

## 6.1 SYNTHETIC EXPERIMENTS

**Data Simulation.** We simulated the data using the Laplace distribution $\exp(-\frac{1}{2}|x|)$, and the mixing matrices are sampled with normally distributed entries with mean 1 and 0.1 standard deviation. The realizations of the observed views are obtained according to the proposed model. In the different scenarios described below we vary the noise distribution.

**Evaluation.** The quality of the mixing matrix estimation is measured with the Amari distance (Amari et al., 1995), which cancels if the estimated matrix differs from the ground truth one up to scale and permutation. More experiments than the one stated below are provided in Appendix D.2

**Noiseless views.** In Figure 3a, we consider a noiseless view setting, where we fixed the dimension to be 100 and we vary the number of shared sources from 10 to 100 in a two view setting. We can see that as soon as the ratio of shared sources to individual sources gets around 1:1 we can recover almost all shared and individual sources compared to the baseline methods which cannot perform well in higher dimensions. The reason of the performance drop of our method is insufficient number of samples for the learning task. See Appendix D.2 for additional experiments in the noisy case.

Table 1: Shared Sources Recovery Performance

| Views | $\sigma = 0.1$ | $\sigma = 0.5$ | $\sigma = 1$ | $\sigma = 2$ |
|---|---|---|---|---|
| 2 | $0.975 \pm 0.002$ | $0.940 \pm 0.003$ | $0.812 \pm 0.003$ | $0.281 \pm 0.01$ |
| 5 | $\mathbf{0.981 \pm 0.001}$ | $0.967 \pm 0.001$ | $0.922 \pm 0.003$ | $0.5 \pm 0.04$ |
| 10 | $0.980 \pm 0.002$ | $\mathbf{0.972 \pm 0.002}$ | $\mathbf{0.95 \pm 0.003}$ | $\mathbf{0.741 \pm 0.08}$ |

**Robustness to model misspecification in a shared response model application.** Here we apply our method to a shared response setting, i.e. no individual sources are available. For this experiment the views have view-specific variance uniformly sampled from $[1, 2]$. Our method shows similar performance to the competitor model MultiViewICA in this special (see Figure 3b) .

**Computing time.** Figure 2 the computing times of response models trained on the same datasets. The experiment was carried out on a local machine using 8 CPU cores in parallel. It seems that MultiViewICA has a similar performance as ShICA-ML but much faster. Our method is faster than ShICA-ML and shows similar mean perfromance as the Infomax method.

**Mean cross correlation (MCC) of shared sources.** We explore how well our method recovers shared sources. We estimate them by taking their average across views and compute the mean cross correlation (MCC) between the estimates and the ground truth. That involves matching estimated with the ground truth components by using the Hungarian algorithm and then computing the mean over all correlations between the aligned pairs. In this experiment, we fixed the total number of sources to be 60 and the shared to be 30. We investigated four cases corresponding to a different noise standard deviation $\sigma = 0.1, 0.5, 1, 2$. and reported the mean MMC and its standard deviation from 50 runs in Table 1. As expected by increasing the number of views we get better estimates of the shared sources, i.e. the MMC score increases.

## 6.2 DATA FUSION OF TRANSCRIPTOME DATA

**Background.** Transcriptome datasets are relevant for the field of genomics. After preprocessing they have the form of random data matrices, where each row correspond to a gene and each column refers to an experiment. Based on these datasets, scientists try to infer gene-gene interactions in the genome.

**Co-regulation Inference.** Since the transcriptome datasets are in the high-dimension-low-sample-size regime (number of genes>number of samples), usually graphical lasso (Friedman et al., 2007) is well-suited for inferring graphical structure from the observed data. More precisely in this application, we want to estimate an undirected graph with nodes referring to the genes and with edges connecting genes with a common regulator.

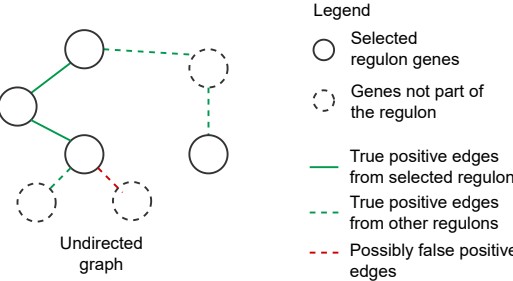

Figure 4: Induced subgraph with vertices from a selected regulon (in the main text denoted by $R$) (not dotted nodes) and its neighbors (dotted nodes). The green edges belong to the ground truth and the red one are possibly false positives.

**Data Integration Task.** To boost the graphical lasso performance, we would like to combine as many experiments as possible. Since the datasets usually come from different labs there is non-biological noise present. Therefore, just pooling the two datasets together for performing downstream tasks can lead to sub-optimal results. The goal of the data integration task is to "denoise" the datasets, such that the transformed data can be used as samples for the graph inference task. In this kind of application usually a dimensionality reduction step is required. This is done by a dimensionality reduction step followed by a feature extracting algorithm, such as IVA-L-SOS, or combined dimensionality reduction method and a feature extractor such as PLS. We follow the latter approach for both, our method and

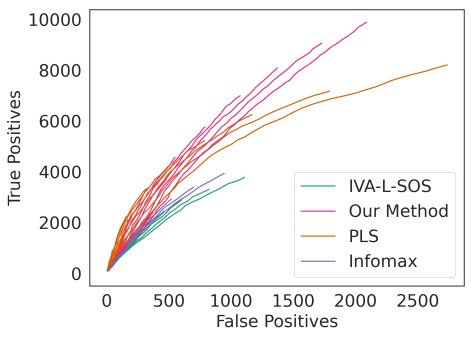
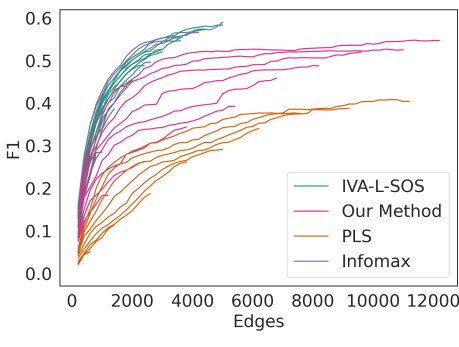

(a) True Positive vs False Positive pairs        (b) F1 score

Figure 5: After the proposed hyperparameter tuning, we compare the top ten EBIC glasso models with our model, PLS, Infomax, and IVA-L-SOS. We order the edges from the selected networks according to their strength. In Figure 5a we count the true positives (y-axis) and possibly false positive edges in the first 100, 200,... edges (x-axis). It seems that our method and PLS outperform the other two methods, and for our method, the true positive/false positive rate increases faster than for PLS. In Figure 5b, we compute the average regulon F1 score (y-axis) from the selected edges (x-axis). Our method seems to perform better than PLS but worse than IVA-L-SOS and Infomax. That might result from the proposed F1 score prioritizing sparse graphs.

Infomax, and directly optimize for unmixing matrices $W_d$ of dimension $c_d \times k_d$ with $c_d \leq k_d$ for $d = 1, \ldots, D$. That means that by minimizing the loss function with respect to possibly non-square $W_1, \ldots, W_D$ we directly extract $c_d$ features, which are then used for the graph inference task.

**Data Assumptions.** We do a one-to-one translation to our proposed model by assuming that each experiment is a noisy linear combination of independent gene pathways and that some pathways are active for both datasets (shared sources).

**Datasets.** In this example, we consider the bacterium *B. subtilis*, for which a very rich collection of the discovered gene-gene interactions are publicly available, which we use as our ground truth model. For this data integration task we use two vast publicly available datasets (Arrieta-Ortiz et al., 2015; Nicolas et al., 2012). Each of the datasets contain gene expression levels of about 4000 genes measured across more than 250 experimental outcomes. For detailed description of the datasets we refer to Appendix C.

**Experiment.** As in most real-life applications, the number of latent sources per dataset is unknown. We treat it as a hyperparameter for each model, i.e., we perform grid search on $\{50, 60, 70, \ldots, 200\}^2$, for the total number of sources for both datasets. Note that for IVA-L-SOS the number of sources in both datasets should coincide. The number of shared sources for our method and PLS varies between $10, 20, 30$, and $40$. We fit 30 graphical lasso models for different penalization parameters on the estimated components. We select the top 10 models by employing a statistical goodness-of-fit measure, called EBIC, for each combination of hyperparameters (see Appendix C for more details). Then, the hyperparameter setting is selected, yielding the best true positive/false positive ratio curves (as the ones shown in Figure 5a). The resulting hyperparameter settings are IVA-L-SOS (130 latent sources), Our Method (50 for dataset 1, 60 for dataset 2, 40 shared sources), PLS (180 for dataset 1, 80 for dataset 2, 10 shared sources), and Infomax (200 for dataset 1, 50 for dataset 2).

**True Positives vs False Positives.** The below-described evaluation is used for our hyperparameter selection. The output graph from the graphical lasso for each pre-processing method is compared to the ground truth one. The evaluation strategy is as follows. For each estimated graph, we order the edges according to their strength. Then we count the true positive and false positive edges in the first $100, 200, \ldots$ edges. Then for each method separately, we select the hyperparameter combination for which the graphical lasso has the best true positive/false positive ratio curves. The results are depicted in Figure 5, where the best models for each method are compared. We can conclude that our model boosts the graphical lasso's performance compared to the others. We also run the graphical lasso on the pooled data without any pre-processing. Surprisingly, the method

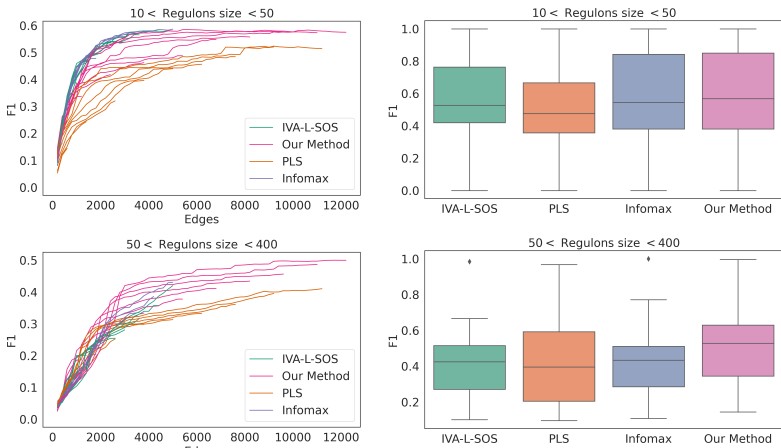

Figure 6: F1 score curves per model (left) and best model F1 regulon performance (right) for two groups of regulons depending on their size (between 10 and 50, and between 50 and 400). In the first group all methods show comparable results. In the second case (regulon size $> 50$) it seems that our method shows better F1 score for the large regulons.

outputs an empty graph, i.e., the goodness-of-fit measure we use evaluates the empty graph as the best model describing the data.

**F1 score.** We are also interested if the output graph resembles the known regulon structure. A regulon is a set of genes that are controlled as a unit by the same regulator. The species of interest has 225 known regulons, which form cliques in our ground truth graph. We compute the precision and recall per regulon from the graphical lasso output in the following way. First, we select a subgraph induced by the regulon genes denoted by $R$ and their neighbors as depicted in Figure 4. The precision is the ratio of the true positive edges and the total number of edges in the selected subgraph. The recall is the ratio of the number of nodes in the biggest connected component with vertices in $R$ and the cardinality of $R$. Note that the recall score does not incorporate any information about the connectivity of the selected subgraph. In Figure 5b we order the edges according to their strength and from the first $100, 200, \ldots$ edges we the F1 score per regulon from the corresponding precision and recall. It seems that IVA-L-SOS and Infomax outperform the other methods. The reason is that the output graphs are sparser in both cases and recover smaller regulons better (most of the regulons have size $< 10$). Due to the overlapping structure of regulons, i.e. larger regulons encompass smaller ones, we want to compare the aggregated F1 scores for the larger regulons. From Figure 6, we can see that our method can recover very large regulons with higher F1 scores than the other methods. In medium size regulon group Infomax, IVA-L-SOS and our method show comparable results.

## 7    DISCUSSION

We proposed a novel noisy linear ICA approach that utilizes the prior knowledge that the different views share information to infer both shared and view-specific sources. Compared to other models from related fields, our model does not assume that all views have the same dimensionality. We provided theoretical guarantees for the identifiability of the model's linear structure and the latent sources in distribution. We adopted a maximum likelihood strategy for estimating the unmixing matrices by maximizing the joint log-likelihood of the observed views. Our empirical results showed that our model performs well in high-dimensional data regimes that often resemble real-life applications. We also suggested a novel strategy for combining transcriptome data and empirically showed that our model improves the performance of a graphical inference model chosen for the particular task. In future work, we would like to address some possible extensions to our model, such as the case when the noise has an arbitrary covariance matrix. Moreover, to state the identifiability results of our model, we assumed that the individual sources from the different views are mutually independent. That might not be the case in real-life applications like the one considered above.

## 8 ETHICS STATEMENT

Our model can be used in many scientific domains such as neuroscience, genomics, physics, medicine, causal discovery, etc., where the proposed model assumptions hold. Therefore, its societal impact is tied to the ethical concerns of the domains it is applied to.

## 9 REPRODUCIBILITY STATEMENT

The code and transcriptome data and ground truth model can be found under `https://anonymous.4open.science/r/shindica-C497/` . We provided detailed proofs of the theorem and other theoretical results stated in the main paper. Furthermore, all synthetic experiments are run multiple times with different random seeds and the reported results are in terms of mean and 95% confidence interval. Additional information regarding the data integration experiment, such as pseudo code of algorithm and explanation of the main steps are provided in the appendix.

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
