# OpenReview forum: "Multi-View Independent Component Analysis with Shared and Individual Sources"
_ICLR.cc/2023/Conference — Submitted to ICLR 2023_

### Official Review · Reviewer_Kgmf · 2022-10-24

**Confidence:** 4
**Correctness:** 3
**Technical Novelty And Significance:** 3
**Empirical Novelty And Significance:** 1
**Recommendation:** 3

**Clarity, Quality, Novelty And Reproducibility:**

(a) The problem of multi-view BSS has not been properly motivated in the introduction. The introduction only provides an introduction on what BSS is, but not the multi-view part.

**Strength And Weaknesses:**

Strengths:
(1) The paper considers the theoretical aspects of multi-view blind source separation
(2) The experiments have properly evaluated against its baselines and other state-of-the-art approaches

Weaknesses:
(3) The approach is very simplistic and lacks any novelty. The entire contribution can pretty much be summarized as Eq.(1) (followed with theoretical guarantees and optimization) which in my opinion does not meet the bar for publication at ICLR.


**Summary Of The Paper:**

The authors proposed a new technique for multi-view blind source separation. The formulation involves a linear summation of a linear approximation of the blind sources from different views (Eq.(1)). The authors then derive a theorem to ensure the identity so that the source signal can be uniquely recovered from the set of mixed signals.


**Summary Of The Review:**

Overall, the paper has a well-evaluated experiment section. The writing of the paper is easy to follow as it is written in dot-point form. But this is non-standard for papers published at ICLR.

I am very familiar with the field of multi-view learning and blind source separation, so I was able to follow the paper. However, I felt that readers who are less familiar with this area will be unable to follow the paper as the introduction (Section 1), Problem Formulation (Section 2), and the theorem in Section 3 are not well written.

I have recommended rejecting the paper because of the combination of (i) clarity of the writing in both motivating the problem and formulating the problem, and (ii) the novelty proposed is incremental which can pretty much be summarized by Eq.(1)

---

> ### Author Response · Authors · 2022-11-11
> **Novelty and Motivation**
>
> We do agree that the contribution is hidden in Eq (1). However, as we now stated in the general response, we believe that the novelty of our method is to address the issues of the other ICA approaches that model only shared sources and to provide a single-step framework for both inferring shared and individual sources that are identifiable. Please refer to the general reply where we addressed the motivation and contribution of our work

---

> ### Author Response · Authors · 2022-11-11
> **Introduction, Problem Formulation (Section 2), and Theorem (Section 3)**
>
> Thank you very much for your feedback. We will provide an updated version of the Introduction soon. In general, we understand that giving explicit and detailed background information about a particular problem would improve the overall understanding of a paper. However, due to the page limit, there is a trade-off how much from the related literature should be explained in detail in the main paper. For our work, we believe that any reader who is familiar with the classical ICA approach should be able to follow the problem formalization and the theoretical results. To make sure that readers who are less familiar with the topic also understand our paper, we will provide additional references.

---

### Official Review · Reviewer_8FCr · 2022-10-24

**Confidence:** 5
**Correctness:** 4
**Technical Novelty And Significance:** 4
**Empirical Novelty And Significance:** 4
**Recommendation:** 8

**Clarity, Quality, Novelty And Reproducibility:**

# Quality
This paper is a pleasure to read, the results of the paper are correct, the model and methods are sound, and the empirical evaluation is convincing. This is a paper of high quality.

# Clarity
This paper is clearly written

# Novelty
The authors study a novel model that is interesting to the community.

# Misc
- In fig.3, I think that Infomax-O is rather Picard-O? If the authors use the picard package then this is definitely not infomax.
- The fonts in the figures are sometimes too small - they should be of the same size as the text

**Strength And Weaknesses:**

# Strengths
- The authors consider a realistic and important problem: indeed, in practice the common assumption of group ICA  that all sources are shared is problematic, and assuming a mix of individual and shared sources is an elegant solution to this problem
- While expected, the identifiability result provides a solid theoretical basis for the method
- The proposed algorithm is sound, as it maximizes a likelihood under orthogonal constraint, as is classical in single subject ICA.
- The experiments are convincing, and clearly show that the proposed algorithm is able to recover the sources, and brings something new on the table in practice.
- The paper is very well written, it was a pleasure to read.

# Weaknesses
- The practical optimization method could probably be improved. Trivialization methods were originally developed for deep learning. Here using these methods destroys some of the nice structure of the problem. For instance we lose the Hessian structure that is used in [Ablin et al 18] and [Richard et al. 20] to get fast algorithms. I think that this is why the proposed method is slower than MultiviewICA (fig.3), while using a similar optimization technique as in [Richard et al. 20] should give a similar runtime.
- The  authors do not release code for the peer review phase, which greatly hinders the reproducibility of this work

**Summary Of The Paper:**

This paper considers the problem of ICA with multiple views, where the goal is to estimate independent sources shared by a group of subject. The originality of this work lies in the addition of individual sources: the data of each subject is a mixture of individual and shared sources, allowing for a more flexible model. The authors demonstrate the identifiability of the proposed model, and propose an algorithm to recover sources and mixing matrices. The authors finally validate their approach on synthetic and genomics problems.

**Summary Of The Review:**

This is a solid paper: the authors study a new model that is relevant to the community, establish some theoretical results about it, and then provide a fast practical algorithm to maximize its likelihood. Finally, the empirical study is convincing. For me, this paper is worthy of acceptance.

---

> ### Author Response · Authors · 2022-11-11
> **Weaknesses**
>
> First of all, thank you so much for the positive feedback about our paper. We are very happy that the reviewer acknowledges the strengths of our method for practical applications.
>
> Regarding the weaknesses:
>
> 1. Thank you for pointing this out. We agree that this is a drawback of our method, which we left for future work. In our paper, the main focus is on addressing the issues of the standard Group ICA approaches as explained in the general response and providing a single-step framework for both inferring shared and individual sources backed up with identifiability results.
>
> 2.  The code will be released in the next couple of days.
>
> Misc: Thank you very much! We will make sure that these changes are implemented in the updated version. The fontsize of the figures are going to be updated upon acceptance

---

### Official Review · Reviewer_g5xN · 2022-10-24

**Confidence:** 4
**Correctness:** 3
**Technical Novelty And Significance:** 2
**Empirical Novelty And Significance:** 2
**Recommendation:** 3

**Clarity, Quality, Novelty And Reproducibility:**

The paper is in general relatively well written and easy to understand, but remains on too high level of abstraction. The task itself is motivated by listing application domains, rather than explaining properly how they are best addressed as multi-view blind source separation problems, and the optimization algorithm is explained only by references to Lezcano-Casado's works and by introducing the notation for the problem formulation. A more formal treatment would be helpful. The technical sections also make the paper feel rather incremental -- every core element is explained by saying how it follows directly from a specific previous work, without really explaining why these choices were made; multi-view models for the basic structure, Richard et al. (2020) for the likelihood expression, and Lezcano-Casado et al. for optimization. The Supplement includes detailed derivation of the mathematical details and suggests that there may be non-trivial elements, but they are not properly communicated in the main paper. Finally, the identifiability proof (which is probably the most interesting result) builds on standard results of Comon (1994) and Davies (2004).

The main problem of the work is that the problem setup itself is poorly motivated. As someone who was worked on MRI data, ICA and multi-view setups I can imagine there are indeed ways of representing MRI data analysis tasks as multi-view BSS problems, but I feel that I would have no hope of understanding the main motivations of this work without having this personal background. For instance, I could not accept the validity of the linearity assumption if not knowing how MRI works and what it measures. Most importantly, however, I do not think you ever give a proper justification on why we should be interested in the specific problem formulation in Section 2. You provide a technical definition and you provide high-level motivation of multi-view settings being interesting in general, but it feels like you are not really verbalising the need for these methods in the context of ICA/BSS.

The empirical experiments are reasonably carried out, but the results do not stand out as strong enough to override the somewhat weak presentation and incremental contribution. It seems that the proposed method works very well on artificial data with high (tens) number of views, but since the paper does not provide good examples of tasks that need this the result remains weak. The transcriptome data, in turn, is a not very easy to understand for typical ICLR readers and while it serves as a nice demonstration it does not add much value.


**Strength And Weaknesses:**

Strengths:
- The theoretical guarantee for identifiability for multi-view learning setups is a valuable contribution

Weaknesses:
- The motivation for multi-view BSS remains poor. The concert example is clearly artificial and the other examples (fMRI etc) are not explained well enough to understand why they correspond to multi-view BSS tasks. The linearity assumption is also poorly motivated, by merely stating that it holds in these applications. It is far from obvious for the reader why fMRI or biological datasets would exhibit prominently linear mixing.
- Core method is poorly explained and seems to be fairly incremental

**Summary Of The Paper:**

The authors extend ICA for multi-view settings, building on similar latent variable models that have been used for modelling correlations between multiple views. This paper extends the formulation for supporting non-gaussian latent distributions to enable identification of individual sources, following the standard ICA principles. The method is demonstrated to outperform competing methods in specific conditions, mostly when there are tens of views.

**Summary Of The Review:**

Technically okay paper that suffers from presentation issues (lack of transparent motivation, algorithmic details on very high level) and incremental contribution. I do not see potential for high impact because the need for the problem remains vague. Empirical experiments are reasonable, but not strong enough to overcome the other problems.

---

> ### Author Response · Authors · 2022-11-11
> **Motivation for the model choice**
>
> Thank you very much for your feedback. Please refer to the general reply for further discussion. We hope it addresses all points raised by the reviewer regarding the motivation of our model.

---

> > ### Comment · Reviewer_g5xN · 2022-11-17
> > **General response**
> >
> > Thank you for the detailed feedback. I am answering all separate elements in this one response, but confirm that I have read all of the isolated remarks.
> >
> > You clarified well the reasoning behind the choices and explained both the need for multi-view setups and the validity of the linearity assumption in these responses, but incorporating all of this information in the manuscript would not be trivial. I agree completely with the linearity assumption being good for these applications and was merely pointing out that you were not describing it properly, whereas regarding the multi-view ICA setup I am still not fully convinced about the practical value. I understand the high-level reasoning, but struggle to figure out how the results would in practice be interpreted or used.
> >
> > I am convinced you would be able to improve the paper by accounting for the feedback, but still feel that even after improving the presentation the paper remains below the publication threshold due to somewhat limited novelty and impact. The paper certainly makes contributions for the field and there are readers especially in the ICA/MRI community that will find it interesting, but the contributions for general learning literature remain too limited. As you said, for example the question of how exactly the optimization is done remains on a level "we did it this way, but it could also be done in some other way".

---

> > > ### Author Response · Authors · 2022-12-07
> > > **Thank you very much for your reply!**
> > >
> > > Thank you very much for your reply! We believe we addressed your questions in the updated manuscript version. We still do not agree that our paper has limited novelty and impact due to the many examples we listed in the [general response](https://openreview.net/forum?id=7WiIzqeqBNL&noteId=Nx2VG1NkOfZ) and the manuscript.
> > >
> > > Regarding the practical value of our work, we gave a real-life example in our work: data integration of omics data. We also believe as the reviewer pointed that the MRI community would be interested in this work.
> > >
> > > Regarding optimization, we never said that the optimization procedure is a contribution of our work. However, we do think that one of our major contributions is that we address an important problem that all shared response models have in common: disregarding the individual sources, which may have a big impact on downstream tasks.

---

> ### Author Response · Authors · 2022-11-11
> **Likelihood Derivation**
>
> We are very sorry for the misunderstanding. With the given reference, i.e., Richard et al. (2020), we meant that we used Richard’s likelihood derivation as a base for our likelihood derivation since the models are related, more precisely, our model generalizes Richard’s. The exact steps are given in the appendix.

---

> ### Author Response · Authors · 2022-11-11
> **Optimization Procedure**
>
> We hope that the following will provide more clarity about the chosen optimization strategy. First, we stated the unconstrained likelihood in Eq. 1.  To simplify the likelihood further, we impose orthogonality constraints on the mixing matrices. Then the loss function becomes as in Eq. 2.  Then, we used the Lezcano-Casado implementation for the orthogonal constraints of the unmixing matrices because of practical reasons (we used pytorch anf the geotorch library for our method implementation). However, we do not claim that this is the only option. One can use other possible approaches as the one cited in the main paper.

---

> ### Author Response · Authors · 2022-11-11
> **Experiments**
>
> "The transcriptome data, in turn, is a not very easy to understand for typical ICLR readers and while it serves as a nice demonstration it does not add much value."
>
> Besides the possible tasks mentioned in the discussion in the general response, we believe that our transcriptome data example is relevant for the field and illustrates how multiview ICA can be utilized for data integration. As mentioned, we are not the first to apply ICA in this context. In many transcriptome applications (as given in the Table in the supplementary), one is interested in learning gene representations used for downstream tasks, such as regulon discovery. Since ICA is such a high-impact method in computational biology, we believe our method gives one possible solution for combining data from different labs in a straightforward way. The chosen downstream task here is co-regulation discovery, performed with graphical lasso.

---

> ### Author Response · Authors · 2022-11-11
> **Identifiability Results**
>
> "the identifiability proof (which is probably the most interesting result) builds on standard results of Comon (1994) and Davies (2004)."
>
> We do agree that this result is built on the classic identifiability results of Comon (1994) and Kagan (1973). However, neither Comon (1994) nor Davies (2004) provides identifiability in distribution of the latent sources and noise in the chosen noisy multiview setting, which we consider a contribution of our work.

---

> ### Author Response · Authors · 2022-11-14
> **Supplement**
>
> "The Supplement includes detailed derivation of the mathematical details and suggests that there may be non-trivial elements, but they are not properly communicated in the main paper."
>
> Thank you for very much for your feedback. We understand that very detailed mathematical derivation in the main paper can benefit the overall understanding. Unfortunately, due to the page limits, there is a tradeoff between what part of the derivations remains in the main text and what part is left in the appendix. We believe that we gave the reader the main steps of our objective function derivations: 1) We stated the unconstrained data log-likelihood 2) We introduced the whitening step and 3) We provided the objective function under orthogonality constraints.  However, if some of the mentioned steps need more clarification, we would happily address them in the main paper.

---

### Official Review · Reviewer_vDv7 · 2022-11-04

**Confidence:** 3
**Clarity, Quality, Novelty And Reproducibility:** 1. Justifications of the assumptions …
**Correctness:** 2
**Technical Novelty And Significance:** 2
**Empirical Novelty And Significance:** 2
**Recommendation:** 5

**Strength And Weaknesses:**

Strength: ICA is a fundamental and important problem, the identifiability of which is also of great importance.

Weakness:

1. the submission focus on the linear case, which limits the applicability of the proposed approach, but I can understand that
2. justification for assumptions are not sufficient
3. experimental results are not sufficient

**Summary Of The Paper:**

This paper studies the problem of multiview linear independent component analysis. The identifiability is analyzed and a constrained joint log-likelihood-based formulation is provided. Experiment results are obtained on real data for a source separation task.

**Summary Of The Review:**

Overall, the reviewer finds the paper studies an important problem, but the experiments need to be improved, and novelty and challenges should be clarified

---

> ### Author Response · Authors · 2022-11-11
> **Assumptions Justification**
>
> "justification for assumptions are not sufficient."
>
> Thank you very much for your comments. In the following, we will justify the identifiability assumptions. To prove the identifiability of the stated model, we require that four assumptions should be satisfied:
>
> 1. The mixing matrices have full-column rank.  This implies that we require that the sources have a minimal representation, i.e. the number of latent sources is minimal, which is a realistic assumption.
>
> 2. The second assumption is additive noise on the sources. It can be interpreted as a measurement error on the device with variance $\sigma^2 A_dA_d^\top$. We choose this setting compared to the $A_ds_d+\epsilon_d$ because, in our case, we get a likelihood in a closed form which is not available in the latter representation. Richard et al. (2020,2021) make a similar assumption for the shared response model setting.
>
> 3. The sources are mutually independent and non-Gaussian. This is a standard ICA assumption (Comon, 1994). Gaussian random variables, called “white” noise represent noise variables, which besides location and scale, do not carry real information.  Thus, if all sources are Gaussian, either they cannot be identified (see, for example,  Proposition 3 (Richard 2020)) or additional assumptions on the variance structure need to be made to assure identifiability (Richard 2021). The non-Gaussian random variables carry information and are identifiable.  This is not a restrictive assumption since the sources in real-life scenarios are often non-Gaussian: fMRI, EEG, and omics data. The fixed mean and variance are also assumptions often adopted in ICA (e.g. (Richard 2021; Hyvarinen, 2000). This results from the fact that the standard literature on ICA does not provide any identifiability guarantees for the true variances and means of the sources.  To illustrate this, we provided a theoretical result in the Supplementary (Theorem A.2), where we do not fix the sources' variances and means. It turns out that we need to make additional assumptions to provide identifiability guarantees for the source distributions.
>
> 4. The measurement error is independent of the latent signal. This is a common assumption in measurement error models known as classical errors. It is a realistic assumption since we usually do not expect the measurement error to influence the true signal and vice versa (Richard 2020,2021; Gresele 2019).

---

> ### Author Response · Authors · 2022-11-11
> **Experiments**
>
> Thank you for your feedback. We will change the color scheme of Figure 5 upon acceptance. Moreover, we appreciate any suggestions on what kind of experiments we can run to improve the empirical justification of our method.

---

> ### Author Response · Authors · 2022-11-11
> **Point 2. from Clarity, Quality, Novelty And Reproducibility**
>
>  We are sorry for the misunderstanding. The used objective function is the data log likelihood of the stated generative model, the derivation of which is based on and extends the derivations of the objective used by Richard (2020).

---

### Author Response · Authors · 2022-12-07
**Reminder**

Dear reviewers,

Thank you once again for your feedback on our work! This is a friendly reminder that we addressed all your concerns in the revised version and we will be pleased to continue the discussion with you.

---

### Decision · Program_Chairs · 2023-01-20

**Decision:**

Reject

**Justification For Why Not Higher Score:**

The assumptions in the paper are not very well justified (while it looks like they can be with a major revision); the experiment results are not very strong.

**Justification For Why Not Lower Score:**

N/A

**Metareview: Summary, Strengths And Weaknesses:**

This paper investigated a new variant of ICA by considering a multiview setting. They analyzed the identifiability for this new model and gave an algorithm. They showed that the proposed method is competitive especially when the number of views is large. While most of the reviewers agree that the results are reasonable, there are some serious concerns about the motivation of the multiview model, and the experiment results. I recommend the authors to carefully revise the paper according to the reviewer suggestions.